# Untreated hypertension in Russian 35-69 year olds – a cross-sectional study

Jakob Petersen[1]*, Anna Kontsevaya[2], Martin McKee[1], Alexander V. Kudryavtsev[3], Sofia Malyutina[4,5], Sarah Cook[6], David A. Leon[1,6]

1 London School of Hygiene & Tropical Medicine, London, United Kingdom, 2 National Research Center for Preventive Medicine, Ministry of Healthcare, Moscow, Russian Federation, 3 Northern State Medical University, Arkhangelsk, Russian Federation, 4 Research Institute of Internal and Preventive Medicine, Branch of Institute of Cytology and Genetics, Siberian Branch of the Russian Academy of Sciences, Novosibirsk, Russian Federation, 5 Novosibirsk State Medical University, Russian Ministry of Health, Novosibirsk, Russian Federation, 6 Department of Community Medicine, UiT The Arctic University of Norway, Tromsø, Norway

* j.petersen@ucl.ac.uk

## Abstract

### Background

The Russian Federation has among the highest rates of cardiovascular disease (CVD) in the world and a high rate of untreated hypertension remains an important risk factor. Understanding who is at greatest risk is important to inform approaches to primary prevention.

### Methods

2,353 hypertensive 35–69 year olds were selected from a population-based study, Know Your Heart, conducted in Arkhangelsk and Novosibirsk, Russian Federation, 2015–2018. The associations between untreated hypertension and a range of co-variates related to socio-demographics, health, and health behaviours were examined.

### Results

The age-standardised prevalence of untreated hypertension was 51.1% (95% CI 47.8–54.5) in males, 28.8% (25.4–32.5) in females, and 40.0% (37.5–42.5) overall. The factors associated with untreated hypertension relative to treated hypertension were younger ages, self-rated general health as very good-excellent, not being obese, no history of CVD events, no evidence of diabetes or chronic kidney disease, and not seeing a primary care doctor in the past year as well as problem drinking for women and working full time, lower education, and smoking for men.

### Conclusion

The study found relatively high prevalence of untreated hypertension, especially, in men. Recent initiatives to strengthen primary care provision and implementation of a general health check programme (dispansarisation) are promising, although further studies should

**Data Availability Statement:** All the data that support the findings of this study are available from Know Your Heart Study (https://knowyourheart. science) subject to scientific approval of a study

protocol. Researchers wishing to get access to the study data can visit the metadata website: https://metadata.knowyourheart.science. The authors of this study had no special access privileges others would not have.

**Funding:** Know Your Heat (KYH) is a component of International Project on Cardiovascular Disease in Russia (IPCDR) and funded by Wellcome Trust Strategic Award [100217; received by DAL], UiT The Arctic University of Norway (UiT), Norwegian Institute of Public Health, and Norwegian Ministry of Health and Care Services. URLs (Accessed 12 November 2019): https://wellcome.ac.uk/ https://en.uit.no/startsida https://www.fhi.no/en/ https://www.regjeringen.no/en/dep/hod/id421/.

**Competing interests:** The authors have declared that no competing interests exist.

evaluate other, potentially more effective strategies tailored to the particular circumstances of this population.

## Introduction

Once detected, hypertension is relatively easy to treat, thereby reducing markedly the risk of complications. Yet many people live for a long time unaware that they have it. A first step in addressing this problem is to determine who, within a population, are most likely to have hypertension that is undetected and untreated. This has been addressed in many populations by studies of the closely associated reasons for non-attendance at general health checks and non-adherence to prescription drugs. These studies point to the importance of knowledge, time, resources, social support, certain beliefs, capability, opportunity, and motivation at the level of the individual and investment in education, healthcare technologies, and healthcare systems at a societal level. Policy responses often focus on the latter, for example through measures to encourage health workers to identify and treat hypertension and to ensure reliable supplies of medicines. Yet, even when well-functioning health systems exist, if those involved fail to recognise the burden on the individual who must change their health seeking and maintaining behaviours to avoid cardiovascular disease (CVD) then failure is likely [1,2]. This makes it necessary to understand the complex barriers and facilitators to diagnosis and treatment and how they vary within populations, and especially those groups that are hard-to-reach.

Studies of non-attendance at general health checks found that those most likely to miss out include the less well educated, less affluent, younger, males, lacking social support, struggling to overcome barriers in terms of geographical and physical access, perceiving themselves not at risk, not yet having a CVD, or smokers [3,4]. Studies of non-adherence to prescription drugs are similar, but also list forgetfulness [5] and factors related to therapy such as adverse drug reactions, prescription error, ineffective and counterfeit drugs, drugs not perceived effective, prescription costs, cumbersome refill prescription systems, beliefs in supplements and alternative remedies lacking scientifically proven efficacy or safety records [6].

In this paper we look at factors associated with untreated hypertension in Russian 35–69 year olds. Russia has very high CVD mortality compared to countries at similar levels of economic development and untreated hypertension is a persisting and important challenge to CVD prevention even though the Russian primary care system is free at the point of use. It has also established a programme of general health checks, offered to the entire adult population, called dispansarisation. This has been expanded progressively since 2013 [7–11].

The aims of the study were to determine the prevalence of untreated hypertension in population samples of two Russian cities, Arkhangelsk and Novosibirsk, and to identify factors associated with untreated hypertension in these two populations.

## Materials and methods

### Population sample

In this study we used data from the Know Your Heart (KYH), a cross-sectional study of cardiovascular structure, function and risk factors in over 4,500 men and women aged 35–69 years from two Russian cities, Arkhangelsk and Novosibirsk, in 2015–2018 [12]. In brief, the study was conducted in three stages: 1) background interview; 2) health check; 3) repeat health check. A list of residential addresses was drawn at random from a population list stratified by

age, sex, district, and city. Trained interviewers visited the addresses and conducted the baseline interview. At the interview, respondents were invited to take part in a health check in a local polyclinic, which included blood pressure measurements. Finally, a stratified random sample of respondents were invited back a year later for a repeat health check. Response rates for the health check component in KYH was 51% in Arkhangelsk and 22% in Novosibirsk based on denominators excluding addresses not found and addresses without at least one person of expected age and gender [12]. 2,353 participants were selected for this study, based on the following inclusion criteria: aged 35–69 years, recorded systolic and diastolic blood pressure, hypertension (blood pressure at or above 140/90 mmHg) or taking antihypertensive medication.

## Screening examination

Blood pressure was measured in KYH using OMRON 705 IT automatic blood pressure monitors (OMRON Healthcare). All devices were calibrated before and after the fieldwork period and no adjustments were needed.

Participants were assigned to different antihypertensive classes based on their systolic and diastolic blood pressure (average of last two out of three measurements) according to the European hypertension treatment guidelines [13] and antihypertensive use. The thresholds for hypertension were a systolic pressure of 140 mmHg or more and/or a diastolic of 90 mmHg or more (office measurement) [13]. Antihypertensive medications were self-reported and comprised those falling within the WHO Anatomical Therapeutic Chemical (ATC) [14] classification system ATC classes, C02, C03, C07, C08, or C09. The main analyses were based on ATC code alone. Sensitivity analyses were conducted using self-reported use of antihypertensive (irrespective of whether a relevant medication was identified among the ATC codes).

To address measurement error (biological variability and measurement method) a random sample of participants stratified by age and sex were invited back a year later for a repeated measurement. The distribution of those who attended the repeated measurement (N = 332) versus those who did not (N = 3,770) is shown in S3 Table. The sample for this part of the study was larger and included all participants and not only those found to have hypertension. To estimate measurement error, only data from participants not on antihypertensive medications and with blood pressures in the normal range (systolic <140 mmHg; diastolic <90 mmHg) at both visits were selected to exclude those that were informed by the study that their blood pressure was too high at the first visit as well as those on treatment. Standard deviation within subjects and within-subject coefficients of variance were calculated [15]. The prevalence of different hypertension status outcomes at the second measurement were calculated and standardised to 2013 European Standard Population for those who were initially found with untreated hypertension.

## Definition of risk factors

Level of education was categorised into four groups: elementary (incomplete secondary, professional no secondary), lower intermediate (complete secondary, professional and secondary), higher intermediate (specialised secondary, incomplete higher), graduate. Self-perceived financial constraints were categorized as: household perceived to be constrained in buying food or clothes versus constrained in buying large domestic appliances versus unconstrained in buying any of the above. Single status was defined as anyone not living with a partner. Self-reported alcohol consumption and alcohol-related behaviours were used to create three categories: Non-drinker past year versus low risk drinkers (score <8) versus high risk drinkers (score 8+) according to WHO alcohol use disorder screening tool [16]. Physical activity was

measured employing the Total physical activity index using the standard classification of inactive versus moderately inactive, moderately active, and active [17]. Diabetes status was ascertained on either self-reported diabetes, self-reported diabetes medication use (ATC A10: insulin or oral antidiabetics) or HbA1c 48+ mmol/mol (>6.5%) [18]. Chronic kidney disease (CKD) status was defined as Glomerular Filtration Rate (eGFR) below 60ml/min/1.73m$^2$ based on serum creatinine [19]. A variable for CVD history comprised any self-reported history of myocardial infarction, heart failure, atrial fibrillation, angina, or stroke events. The presence of depression was included in the analyses because it has been associated with CVD mortality, and specifically in Eastern European countries [20]. Here it was defined as having a score at and above 5 on the PHQ-9 instrument [21]. Attendance at a general health check was ascertained using a question about having attended the dispansarisation programme since its relaunch in 2013; this was used in the descriptive analysis [7]. Economic activity was defined according to responses to routed questions on retirement (baseline questionnaire item A11), regular paid work (A12), and other activity (A14). Hypertension knowledge was tested with a question about whether hypertension is believed to always, sometimes, or never be associated with symptoms.

## Statistical analysis

The prevalence of untreated hypertension was estimated and standardized by age and sex to the European Standard Population [22]. Gender-specific multivariate logistic regression models of untreated versus treated hypertension in 35–69 year olds adjusting for age or age and CVD history were fitted using Stata 15 [23]. A range of co-variates was included as potential factors associated with untreated hypertension, i.e. education, marital status, body mass index, alcohol consumption, depression (PHQ-9), and whether a primary care doctor was visited in the past year.

## Ethical approval

The study complied with the 1964 Helsinki declaration and its later amendments and ethical approval was received from the ethics committees of London School of Hygiene and Tropical Medicine (approval number 8808), Novosibirsk State Medical University (approval number 75; 21 May 2015), the Institute of Preventative Medicine (approval received 26 December 2014), Novosibirsk and the Northern State Medical University, Arkhangelsk (approval number 01/01-15; 27 January 2015). Informed written consent was obtained from all individual participants included in the study.

## Results

The characteristics of the study participants (N = 2,353) are shown in S1 and S2 Tables. The prevalence by gender of the different hypertensive categories, including normotensives, can be found at Fig 1.

To address measurement error, a random sample of 332 participants (178 females and 154 males) were invited back a year later for a repeated measurement (Fig 2). 109 participants, normotensive at both visits, were selected. The mean duration between the two measurements was 359 days (SD 22). Standard deviation within subjects were 5.8 mmHg and 3.5 mmHg for systolic and diastolic measurements, respectively. The within-subject coefficients of variance for the same measurements were 5.0% and 4.6%.

For those in the repeated measurement sub-sample (N = 332) found to have untreated hypertension at the first visit, the most common trajectory for men was to be classified as having untreated hypertension again (55%; 95% CI 42–68) and only 17% (9–30) were on

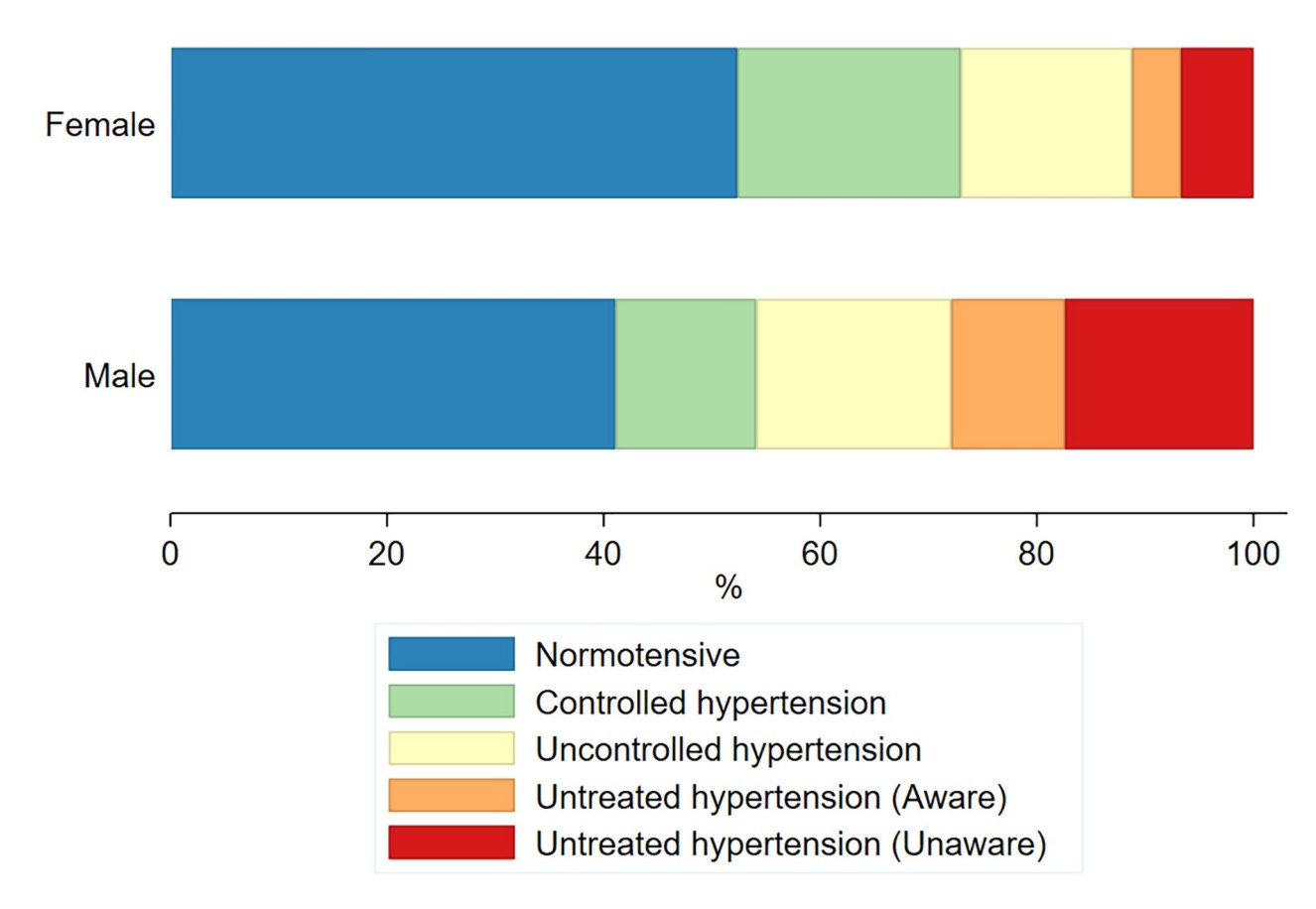

**Fig 1. Age-standardised prevalence (%) of hypertensive categories among 35–69 year olds by gender.**

antihypertensive medications a year later. For women, the same proportions were 37% (22–54) and 37% (22–54).

The prevalence of untreated hypertension among those with hypertension was 51.1% (95% CI 47.8–54.5) for males, 28.8% (25.4–32.5) for females, and 40.0% (37.5–42.5) overall (S3 Table).

The factors associated with untreated hypertension relative to treated hypertension were younger ages, self-rated general health as very good-excellent, not being obese, no history of CVD events, no evidence of diabetes or CKD, and not seeing a primary care doctor in the past year as well as problem drinking for women and working full time, lower education, and smoking for men (Tables 1 and 2). These effects held even when, in addition, adjusting for CVD history.

For women, hypertension awareness was 85% among those with treated hypertension and 40% among those untreated (S1 Table). For men, the same proportions were similar, 85% and 38%, respectively (S2 Table).

For women, general health check attendance was 57% in those with untreated hypertension and 50% in those with treated hypertension (S1 Table). For men, 33% and 29% (S2 Table).

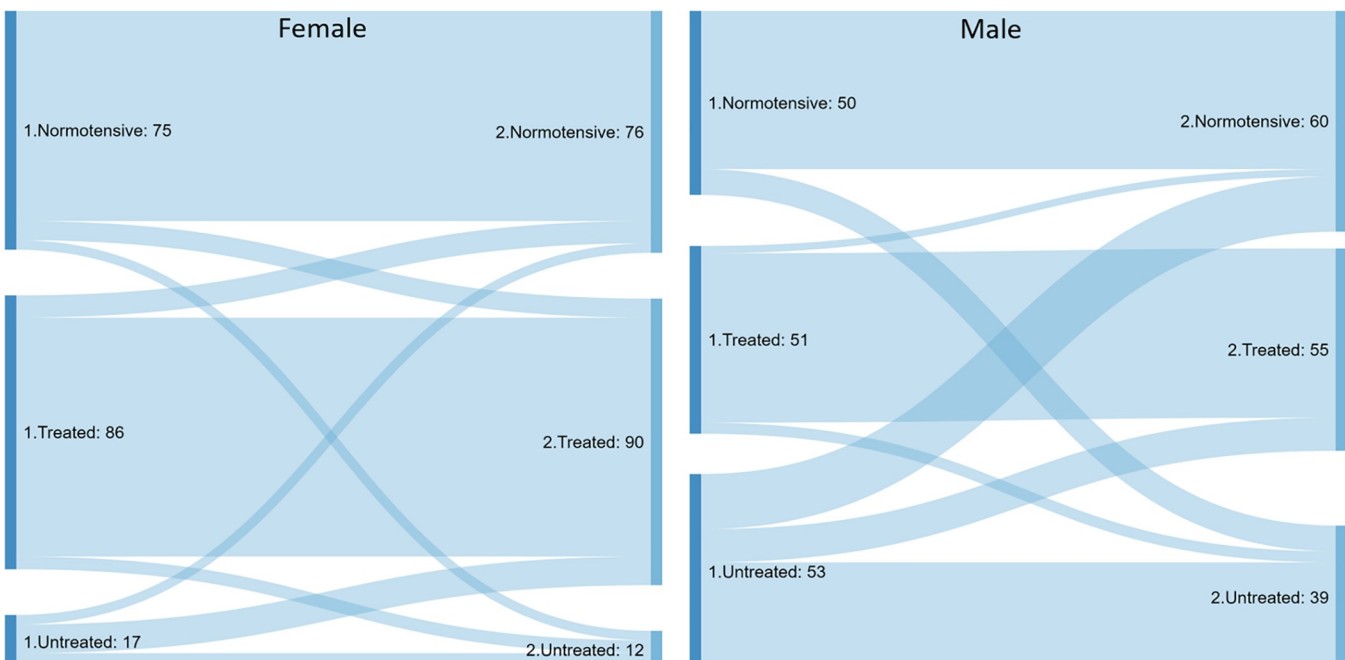

**Fig 2.** Hypertension status flows between first and repeated measurement for a subset of female (left panel; N = 178) and male (right; N = 154) 35–69 year olds.

The majority of respondents was unaware that hypertension was an asymptomatic condition (S1 and S2 Tables). The most 'knowledgeable' were untreated men (17% of men knew the correct answer) and the least were treated women (8%).

## Discussion

Prevalence of untreated hypertension among the 35–69 year olds in this study was 51% in males, 29% in females, and 40% overall. Similar levels were found in the concurrent ESSE-RF Study carried out in a population-based sample of 25–64 year olds in four other Russian regions, i.e. 58% in men, 29% in women, and 47% overall [24]. In comparison, a study with data from 123 nationally representative surveys of 40–79 year olds in 12 high income countries [25] found that prevalence of untreated hypertension in males ranged from 19% in Canada to 61% in Ireland. For females, from 20% in Germany and the US to 50% in Ireland. The KYH study population would therefore fall into the mid-range for males and towards the best performing end for females. The prevalence in males was on par with Italy (44%), Australia (45%), Finland (45%), New Zealand (45%), UK (45%), Japan (48%), and Spain (49%). For women, it was on par with South Korea (26%), Germany (20%), and the US (20%). Common to the best performing countries, were that national clinical guidelines recommend treatment of blood pressures at and above 140/90 mmHg and a working general health check programme. Russian health professionals should follow European treatment guidelines, initiating treatment at blood pressures at and above 140/90 mmHg. The dispansarisation programme, expanded in 2013, is central to preventive efforts. At the time of data collection, in 2015–2018, among those with untreated hypertension 50% of women and 30% of men had attended a general health check since 2013 versus 57%/33% among those with treated hypertension. This indicates that the dispansarisation programme still has some way to go in terms of coverage. It should be noted that legislation enacted in 2019 aims to strengthen the dispansarisation

**Table 1. Logistic regression models of untreated versus treated hypertension among female participants (N = 1,265).**

| Characteristic | Level | AOR age | P-value | 95% CI | AOR age/CVD | P-value | 95% CI |
|---|---|---|---|---|---|---|---|
| CVD history | No | Ref | | | | | |
| | Yes | **0.26** | < .001 | 0.18–0.38 | | | |
| Age group | 35–49 yr | Ref | | | Ref | | |
| | 50–59 yr | **0.58** | .003 | 0.41–0.83 | **0.67** | .031 | 0.46–0.96 |
| | 60–69 yr | **0.33** | < .001 | 0.23–0.47 | **0.46** | < .001 | 0.32–0.66 |
| Education | Elementary | Ref | | | Ref | | |
| | Lower intermediate | 1.35 | .397 | 0.67–2.72 | 1.47 | .290 | 0.72–2.99 |
| | Higher intermediate | 1.38 | .329 | 0.72–2.65 | 1.48 | .244 | 0.76–2.87 |
| | Graduate | 1.63 | .148 | 0.84–3.17 | 1.63 | .157 | 0.83–3.20 |
| Economic activity | Retired | Ref | | | Ref | | |
| | Paid work | 1.57 | .073 | 0.96–2.57 | 1.45 | .137 | 0.89–2.38 |
| | Looking after home | 0.91 | .701 | 0.58–1.44 | 0.95 | .818 | 0.59–1.51 |
| | Unemployed | 0.78 | .711 | 0.20–2.97 | 0.82 | .780 | 0.21–3.23 |
| | Other | 2.16 | .326 | 0.46–10.1 | 2.43 | .276 | 0.49–12.0 |
| Household income | Constrained | Ref | | | Ref | | |
| | Intermediary | 0.90 | .543 | 0.64–1.26 | 0.83 | .304 | 0.59–1.18 |
| | Rel. unconstrained | 1.12 | .555 | 0.76–1.66 | 1.00 | .976 | 0.68–1.50 |
| Single | No | Ref | | | Ref | | |
| | Yes | 0.85 | .265 | 0.65–1.13 | 0.92 | .566 | 0.69–1.22 |
| Smoking | No | Ref | | | Ref | | |
| | Yes | 1.21 | .312 | 0.84–1.74 | 1.23 | .270 | 0.85–1.79 |
| Alcohol use disorder | Non-drinker past year | Ref | | | Ref | | |
| | Low (AUDIT<8) | 1.15 | .494 | 0.77–1.72 | 1.09 | .684 | 0.72–1.64 |
| | High (AUDIT 8+) | **3.55** | .002 | 1.61–7.84 | **3.04** | .007 | 1.36–6.77 |
| Physical activity | Inactive | Ref | | | Ref | | |
| | Moderately inactive | 1.34 | .446 | 0.63–2.83 | 1.49 | .306 | 0.70–3.18 |
| | Moderately active | 1.25 | .500 | 0.65–2.40 | 1.39 | .328 | 0.72–2.67 |
| | Active | 1.08 | .816 | 0.54–2.16 | 1.15 | .697 | 0.57–2.30 |
| Self-rated general health | Poor/fair/good | Ref | | | Ref | | |
| | Very good/excellent | **2.20** | < .001 | 1.65–2.94 | **1.97** | < .001 | 1.47–2.64 |
| Body Mass Index | Under/Normal (<25) | Ref | | | Ref | | |
| | Overweight (25–29) | 1.00 | .988 | 0.67–1.47 | 0.95 | .800 | 0.64–1.41 |
| | Obese (30–34) | **0.60** | .015 | 0.40–0.91 | **0.60** | .019 | 0.40–0.92 |
| | Very obese (35+) | **0.35** | < .001 | 0.21–0.58 | **0.35** | < .001 | 0.21–0.58 |
| Diabetic | No | Ref | | | Ref | | |
| | Yes | **0.36** | < .001 | 0.22–0.59 | **0.38** | < .001 | 0.23–0.63 |
| CKD | No | Ref | | | Ref | | |
| | Yes | 0.65 | .187 | 0.35–1.23 | 0.70 | .284 | 0.37–1.34 |
| Depression (PHQ-9) | No | Ref | | | Ref | | |
| | Yes | **0.67** | .006 | 0.51–0.89 | 0.76 | .060 | 0.57–1.01 |
| Seen primary care doctor past year | No | Ref | | | Ref | | |
| | Yes | **0.29** | < .001 | 0.21–0.39 | **0.32** | < .001 | 0.23–0.43 |

Age- and CVD history-adjusted odds ratios (AOR) with 95% confidence intervals.

programme further by offering it yearly to those aged 40 years and above, introducing an online appointment system for users and opening evening and Saturday appointments [11].

**Table 2. Logistic regression models of untreated versus treated hypertension among male participants (N = 1,088).**

| Characteristic | Level | AOR age | P-value | 95% CI | AOR age/CVD | P-value | 95% CI |
|---|---|---|---|---|---|---|---|
| CVD history | No | Ref | | | | | |
| | Yes | **0.23** | < .001 | 0.17–0.32 | | | |
| Age group | 35–49 yr | Ref | | | Ref | | |
| | 50–59 yr | **0.48** | < .001 | 0.35–0.67 | **0.57** | .002 | 0.40–0.81 |
| | 60–69 yr | **0.29** | < .001 | 0.21–0.40 | **0.41** | < .001 | 0.30–0.58 |
| Education | Elementary | Ref | | | Ref | | |
| | Lower intermediate | 0.68 | .104 | 0.42–1.08 | 0.62 | .062 | 0.38–1.02 |
| | Higher intermediate | 0.65 | .061 | 0.42–1.02 | **0.59** | .032 | 0.37–0.96 |
| | Graduate | **0.59** | .022 | 0.38–0.93 | **0.50** | .004 | 0.31–0.81 |
| Economic activity | Retired | Ref | | | Ref | | |
| | Paid work | **1.92** | .001 | 1.31–2.81 | **1.64** | .014 | 1.10–2.45 |
| | Looking after home | 1.05 | .895 | 0.52–2.12 | 1.16 | .694 | 0.55–2.45 |
| | Unemployed | 1.99 | .083 | 0.91–4.33 | 2.22 | .057 | 0.98–5.04 |
| | Other | 1.33 | .486 | 0.60–2.95 | 1.25 | .604 | 0.54–2.85 |
| Household income | Constrained | Ref | | | | | |
| | Intermediary | 1.04 | .819 | 0.74–1.46 | 0.88 | .497 | 0.62–1.26 |
| | Rel. unconstrained | 1.13 | .522 | 0.78–1.62 | 0.84 | .376 | 0.57–1.23 |
| Single | No | Ref | | | Ref | | |
| | Yes | 1.40 | .056 | 0.99–1.98 | 1.40 | .068 | 0.98–2.00 |
| Smoking | No | Ref | | | Ref | | |
| | Yes | **1.39** | .012 | 1.07–1.80 | **1.43** | .010 | 1.09–1.87 |
| Alcohol use disorder | Non-drinker past year | Ref | | | Ref | | |
| | Low (AUDIT<8) | 1.21 | .198 | 0.90–1.63 | 1.13 | .435 | 0.83–1.54 |
| | High (AUDIT 8+) | 1.33 | .073 | 0.97–1.82 | 1.17 | .342 | 0.85–1.63 |
| Physical activity | Inactive | Ref | | | Ref | | |
| | Moderately inactive | 0.60 | .148 | 0.30–1.20 | 0.64 | .223 | 0.31–1.31 |
| | Moderately active | 0.67 | .199 | 0.36–1.23 | 0.70 | .266 | 0.37–1.32 |
| | Active | 1.05 | .886 | 0.56–1.96 | 1.02 | .944 | 0.53–1.96 |
| Self-rated general health | Poor/fair/good | Ref | | | Ref | | |
| | Very good/excellent | **2.18** | < .001 | 1.69–2.81 | **1.77** | < .001 | 1.35–2.30 |
| Body Mass Index | Under/Normal (<25) | Ref | | | Ref | | |
| | Overweight (25–29) | **0.58** | .001 | 0.42–0.80 | **0.58** | .002 | 0.41–0.81 |
| | Obese (30–34) | **0.32** | < .001 | 0.22–0.47 | **0.31** | < .001 | 0.21–0.47 |
| | Very obese (35+) | **0.20** | < .001 | 0.11–0.35 | **0.22** | < .001 | 0.12–0.40 |
| Diabetic | No | Ref | | | Ref | | |
| | Yes | **0.26** | < .001 | 0.16–0.41 | **0.30** | < .001 | 0.19–0.49 |
| CKD | No | Ref | | | Ref | | |
| | Yes | **0.35** | .007 | 0.17–0.75 | **0.44** | .040 | 0.20–0.96 |
| Depression (PHQ-9) | No | Ref | | | Ref | | |
| | Yes | **0.69** | .011 | 0.52–0.92 | 0.86 | .337 | 0.64–1.17 |
| Seen primary care doctor past year | No | Ref | | | Ref | | |
| | Yes | **0.35** | < .001 | 0.27–0.46 | **0.39** | < .001 | 0.30–0.51 |

Age- and CVD history-adjusted odds ratios (AOR) with 95% confidence intervals.

Untreated hypertension was found to be associated with younger ages, self-rated general health as very good-excellent, not being obese, no history of CVD events, no evidence of

diabetes or CKD, and not seeing a primary care doctor in the past year as well as problem drinking for women and working full time, lower education, and smoking for men.

These findings are largely consistent with the literature on non-attendance to participatory health interventions and drug non-adherence [3,4,6,26], although financial constraints were not found statistically significant or important. The ESSE-RF Study in Russia found younger ages and absence of CVD associated with untreated hypertension [24]. Other factors were rural residence and higher lipids in men and higher heart rates in women.

Interestingly, working full time for men was associated with untreated hypertension even after adjusting for age and CVD history. This suggests that not having sufficient time to navigate the healthcare system to get a health check and appropriate treatment could be a barrier, especially, for men. This finding points to a need to undertake more studies on how to lower the barriers for full time workers. To this end, under legislation enacted in 2018, workers are now entitled to paid leave when attending the dispansarisation health checks [27].

Of those with untreated hypertension, as many as 40% of women and 38% of men reported to have been diagnosed with hypertension at some point (hypertension awareness). Only 14% of women with untreated hypertension and 17% of men knew that hypertension was an asymptomatic condition. A study of first time attenders to a cardiology clinic in Moscow with six month follow-up similarly found absence of symptoms (73%) to be the most common reason given for non-adherence [5]. Less commonly given reasons were forgetfulness (31%), out-of-pockets costs (22%), non-eligibility of re-imbursement schemes (14%), side-effects (10%), treatment courses (7%), and polypharmacy (5%). This points to particular issues around health knowledge and medication adherence that should be studied further. Other studies have found the misconception that hypertension is a condition brought on temporarily by stress and that antihypertensive medications no longer need be taken if the feelings of stress pass [28].

For those with untreated hypertension at the first visit, the most common trajectory for men was to be diagnosed with untreated hypertension again (55%; 95% CI 42–68), 17% were on antihypertensive medications (9–30), and only 3.7% (0.9–14) with controlled hypertension a year later. For women, the same proportions were 37% (22–54), 37% (22–55), and 34% (19–53) (Fig 1). Although these findings were based on a very small number of participants, they do suggest that women are more likely than men to act upon health advice when informed about a preventable health issue such as untreated hypertension. This finding could inform encounters in general practice when prompting patients to take part in health checks as well as in public health campaigns tailored to male audiences.

Detection, treatment, and control of hypertension have improved internationally since the 1980s in high-income countries, yet control has stagnated since the mid- to late 2000s, pointing to a need for innovative strategies or interventions should be considered in addition to the existing measures. New interventions may exploit new technologies [25], raise awareness through large pragmatic studies such as the May Measurement Month project [29] where participants can contribute to international research, or explore new effective settings for outreach such as the example of the Los Angeles black barbershop trial [30]. At a more strategic level, there are examples of interventions that have achieved good results by offering comprehensive and contextualised care involving networks of non-physician health workers [31]. A recent review of the effectiveness of blood pressure control strategies with 100 articles concluded that multilevel, multicomponent strategies followed by patient-level strategies were the most efficient [32]. Examples of effective strategies included team-based care strategies and strategies providing training for home monitory of blood pressure and health coaching in primary care and community settings. The review also concluded that trial evidence on strategies involving electronic decision support systems is still sparse and should be evaluated further.

## Limitations

Sampling bias introduced by non-response was assessed by comparing the realised sample against data from the Russian Census 2010 on age, gender, and higher education attainment [12]. Overall, the realised sample for the health check was close to equity, with a ratio of 0.99 (95% CI 0.93–1.06) for Arkhangelsk and 1.26 (1.17–1.34) for Novosibirsk. It cannot be excluded that non-responders could have been less likely to seek healthcare and adhere to medical advice than responders. Among those that did not attend the health check, there were relatively fewer with CVD and hypertension awareness at the baseline compared to those that did attend [12]. The true prevalence of untreated hypertension is therefore likely to be even higher than estimated in this study.

In addition to non-response biases, the study will by design have excluded institutionalised individuals, individuals with no fixed abode, and individuals too ill to be interviewed or to take part in the health check carried out at a local clinic, and as such may not be fully representative of groups that currently are hard to reach with public health interventions. This is a limitation that should be borne in mind when interpreting the results of the study.

The definition of hypertension used in this study relied on a precise measurement of blood pressure at a single time point. If there were a large measurement error associated with blood pressure measurement either due to biological variability or measurement method, this could affect the outcome variable and potentially dilute the purported associations with risk factors [33]. The within-individual standard variation was found to be relatively modest among individuals measured again a year later. Only individuals normotensive at both visits were included in this part of the study to, as far as possible, exclude factors associated with treatment as well as those who prompted by the results of the first visit could have sought further care or ameliorated their health behaviours accordingly.

Identification of those treated for hypertension was based on self-reported medication data but not every participant may have been aware of the indication for each medicine reported. As a sensitivity analysis, prevalence of untreated hypertension was calculated using a question on whether antihypertensive medications were taken, with similar results.

## Conclusion

The study found a relatively high prevalence of untreated hypertension, especially, in men. Initiatives to strengthen the primary care provision and the dispansarisation general health check programme making it yearly for those aged 40 years and above, extending hours, introducing Saturday appointments as well as paid leave for those in full time work to attend are promising. Further studies should however investigate whether more effective strategies could be designed to accommodate population needs in terms of detection and adherence to antihypertensive medications.

## Supporting information

**S1 Table. Characteristics of female study participants: Treated versus untreated hypertension (N = 1,265).**
(DOCX)

**S2 Table. Characteristics of male study participants: Treated versus untreated hypertension (N = 1,088).**
(DOCX)

**S3 Table. Prevalence (%) of untreated hypertension among hypertensive 35–69 year olds.**
(DOCX)

**S4 Table. Characteristics of study participants with any hypertension status incl. normotensive (N = 4,102): Without repeated measurement versus with repeated measurement.** (DOCX)

## Author Contributions

**Conceptualization:** Jakob Petersen, Anna Kontsevaya, Martin McKee, Alexander V. Kudryavtsev, Sofia Malyutina, Sarah Cook, David A. Leon.

**Data curation:** Jakob Petersen.

**Formal analysis:** Jakob Petersen.

**Funding acquisition:** Martin McKee, Sofia Malyutina, David A. Leon.

**Writing – original draft:** Jakob Petersen.

**Writing – review & editing:** Jakob Petersen, Anna Kontsevaya, Martin McKee, Alexander V. Kudryavtsev, Sofia Malyutina, Sarah Cook, David A. Leon.

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
