## [Decision Letter · Decision Letter 0]

17 Feb 2020

PONE-D-19-34971

Untreated hypertension in Russian 35-69 year olds – a cross-sectional study

PLOS ONE

Dear Dr. PETERSEN,

Thank you for submitting your manuscript to PLOS ONE. After careful consideration, we feel that it has merit but does not fully meet PLOS ONE’s publication criteria as it currently stands. Therefore, we invite you to submit a revised version of the manuscript that addresses the points raised during the review process.

I agree with both reviewers your manuscript can benefit from subheadings and better structuring of the Methods section.

Also provide more details on study population selection and expand the discussion section on how to make inferences about generalizability of odds ratios given the difference in participation rates between the two urban populations. Are these still valid or would the estimation of odds ratios be affected due to the selection process? 

Reviewer 2 makes an important point about including psychosocial CVD risk factors such as depression, anxiety and stress. These variables should be included in the multivariable analysis when available.

Please also provide results from a logistic regression model including all variables (not only adjusted for age and CVD, but adjusting with all other variables) and test for interaction by gender (within one comprehensive model, not separated by gender). 

We would appreciate receiving your revised manuscript by Apr 02 2020 11:59PM. To enhance the reproducibility of your results, we recommend that if applicable you deposit your laboratory protocols in protocols.io, where a protocol can be assigned its own identifier (DOI) such that it can be cited independently in the future. For instructions see: http://journals.plos.org/plosone/s/submission-guidelines#loc-laboratory-protocols

We look forward to receiving your revised manuscript.

Kind regards,

Bart Ferket

Academic Editor

PLOS ONE

Journal Requirements:

2. In ethics statement in the manuscript and in the online submission form, please provide additional information about what type of informed consent you obtained (for instance, written or verbal, and if verbal, how it was documented and witnessed).

Reviewers' comments:

Reviewer's Responses to Questions

**Comments to the Author**

1. Is the manuscript technically sound, and do the data support the conclusions?

Reviewer #1: Partly

Reviewer #2: Partly

2. Has the statistical analysis been performed appropriately and rigorously? 

Reviewer #1: Yes

Reviewer #2: I Don't Know

3. Have the authors made all data underlying the findings in their manuscript fully available?

Reviewer #1: No

Reviewer #2: Yes

4. Is the manuscript presented in an intelligible fashion and written in standard English?

Reviewer #1: No

Reviewer #2: Yes

5. Review Comments to the Author

Reviewer #1: The aim of the study is not clearly stated. I just assume it was (1) to determine the prevalence of untreated hypertension in population samples of Arkhangelsk and Novosibirsk and (2) to identify factors to be associated with untreated hypertension in the two population samples.

Section Materials and methods should be clearly structured having the following subsections:

Population – this should also include how the individuals were selected

Screening examination – should be briefly described, including blood pressure measurement

Definition of risk factors, incl. hypertension. The definition of hypertension used on lines 117 and 118 does not seem to include values equal to 140/90 mmHg, which is not in agreement with the quoted European guidelines on hypertension.

Statistical analysis

In conclusion, it is an important study which has to be published, however, the current version needs a major revision.

Reviewer #2: The submitted manuscript assesses the rate of untreated hypertension based on the «Know Your Heart» study and is carried out in two Russian cities, namely Novosibirsk and Arkhangelsk. The aim was also to provide further insights into the country-specific factors contributing to the fact that far too many hypertensive men and women currently remain untreated. However, there are a number of points, which need clarifications and revision.

1) The Materials and Methods section is not enough conclusive. It is difficult to understand the methodology of the study. Although authors provided a reference to their comprehensive paper on the «Know Your Heart» study design [Cook S, Malyutina S, Kudryavtsev AV, et al. Know Your Heart: Rationale, design and conduct of a cross-sectional study of cardiovascular structure, function and risk factors in 4500 men and women aged 35-69 years from two Russian cities, 2015-18. Wellcome Open Res. 2018;3:67.], it is not enough. I recommend to give in the current paper at least basic explanations on how the study population was selected.

2) The description of statistical methods is very scarce; they are presented literally in a couple of sentences and are not allocated in a separate subsection.

3) The robustness of results seems doubtful due to a low response rate. Moreover, the response rate in Novosibirsk, which is a developed city with a population of more than 1.5 million people, was more than two times lower than in Arkhangelsk (22% vs 51%, respectively). This issue must be addressed in some way.

4) Within the scope of the studied problems, the psychosocial CVD risk factors such as depression, anxiety and stress are shown to be of great importance for the Russian population. These factors serve as barriers for traditional risk factors modification and CVD treatment, including hypertension treatment, and they are also known to independently affect long-term health outcomes in CVD patients. According to the design paper, the “Know Your Heart” study encompassed assessment of anxiety and depression (using validated questionnaires), as well as assessment of self-reported social support. It is not clear, why these data were not used in the multivariate logistic regression models of untreated vs treated hypertension in this paper. The inclusion of these data into the models may provide more accurate and comprehensive information on factors, associated with lack of antihypertensive treatment, specifically relevant for the Russian population.

5) I would recommend avoiding inappropriate slang expressions like antihypertensives instead of antihypertensive drugs or medicines or treatment.

Given all of the above, this article needs a major revision before publication.

6. PLOS authors have the option to publish the peer review history of their article (what does this mean?). If published, this will include your full peer review and any attached files.

Reviewer #1: Yes: Renata Cifkova

Reviewer #2: No

---

## [Author Response · Author response to Decision Letter 0]

6 May 2020

PLOS ONE

Re: Untreated hypertension in Russian 35-69 year olds – a cross-sectional study (PONE-D-19-34971)

Dear Dr Bart Ferket,

Please find our response to the reviewers’ comments below. We found the comments constructive and feel that the process has strengthened the paper.

Many regards and on behalf of the author team,

Jakob Petersen

London, 03 March 2020

Attachments: Response to Reviewers, Revised Manuscript with Track Changes, Manuscript

Editor’s comments

I agree with both reviewers your manuscript can benefit from subheadings and better structuring of the Methods section.

Response:

We agree that the structure and content suggested by the reviewers will make the Methods section much clearer. We have now restructured and expanded it accordingly.

Also provide more details on study population selection and expand the discussion section on how to make inferences about generalizability of odds ratios given the difference in participation rates between the two urban populations. Are these still valid or would the estimation of odds ratios be affected due to the selection process? 

Response:

We agree that the low response rate, especially in one of the two cities, could lead to sampling bias. We currently discuss this in the Limitations sub-section, where we refer to analyses carried out by Cook et al. (2018) in our earlier paper on the study methodology. Cook et al. compared age, sex, and higher education attainment of respondents with the Russian 2010 Census population of each city. The results showed that the sample attending for the health check was almost the same as that in the Census, with a ratio of 0.99 (95% CI 0.93-1.06) for Arkhangelsk and 1.26 (1.17-1.34) for Novosibirsk. 

We also note that our prevalence estimates of untreated hypertension were very close to the concurrent ESSE-RF Study of 25-64 year olds in four other Russian regions (Balanova et al. 2019).

We feel that the relatively low response rate needs to be acknowledged when interpreting the results. Based on the comparison with the Russian Census population we have no good reason to conclude that the respondents differed markedly in terms of age, sex, and higher education attainment. 

Reviewer 2 makes an important point about including psychosocial CVD risk factors such as depression, anxiety and stress. These variables should be included in the multivariable analysis when available.

Response:

We agree and now include a reference in support of this suggestion and have included depression status (PHQ-9 instrument) in the analyses. Given that the constructs for depression, anxiety, and stress are related, we suggest that only the most severe of the three, i.e. depression, is included.

Please also provide results from a logistic regression model including all variables (not only adjusted for age and CVD, but adjusting with all other variables) and test for interaction by gender (within one comprehensive model, not separated by gender). 

Response:

We present the joint model for both genders below (age- and sex-adjusted and fully adjusted odds ratios). No significant interactions between age group and gender were found. This was tested with a likelihood ratio test in the full model without an interaction term versus the full model with an interaction term. 

It is well known that men and women differ in health behaviours, for many reasons including gender norms and expectations and targeted marketing of unhealthy products, and we suggest to keep the gender-specific regression tables.

Table Logistic regression models of untreated versus treated hypertension. Age- and fully adjusted odds ratios (AOR) with 95% confidence intervals.

Characteristic Level AOR age/sex P-value 95% CI AOR all P-value 95% CI

Gender Female Ref Ref 

 Male 2.74 <.001 2.28-3.29 1.67 <.001 1.28-2.19

CVD history No Ref Ref 

 Yes 0.24 <.001 0.19-0.31 0.31 <.001 0.24-0.40

Age group 35-49 yr Ref Ref 

 50-59 yr 0.53 <.001 0.41-0.67 0.71 .034 0.51-0.97

 60-69 yr 0.31 <.001 0.24-0.39 0.63 .017 0.43-0.92

Education Elementary Ref Ref 

 Lower intermediate 0.84 .368 0.58-1.23 0.85 .450 0.56-1.30

 Higher intermediate 0.83 .290 0.58-1.18 0.80 .265 0.54-1.19

 Graduate 0.84 .336 0.59-1.20 0.71 .106 0.47-1.07

Economic activity Retired Ref Ref 

 Paid work 1.71 <.001 1.28-2.29 1.39 .051 1.00-1.94

 Looking after home 0.96 .833 0.66-1.40 0.89 .599 0.58-1.37

 Unemployed 1.41 .286 0.75-2.67 1.76 .132 0.84-3.67

 Other 1.38 .369 0.68-2.81 0.92 .848 0.41-2.09

Household income Constrained Ref Ref 

 Intermediary 0.97 .801 0.76-1.23 0.83 .181 0.63-1.09

 Rel. unconstrained 1.11 .437 0.85-1.45 0.87 .393 0.63-1.20

Single No Ref Ref 

 Yes 1.04 .734 0.84-1.29 1.02 .876 0.80-1.30

Smoking No Ref Ref 

 Yes 1.32 .009 1.07-1.63 1.14 .305 0.89-1.46

Alcohol use disorder Non-drinker past year Ref Ref 

 Low (AUDIT<8) 1.21 .108 0.96-1.54 0.93 .619 0.68-1.26

 High (AUDIT 8+) 1.49 .006 1.12-1.98 1.00 .996 0.68-1.47

Physical activity Inactive Ref Ref 

 Moderately inactive 0.87 .569 0.53-1.42 1.13 .668 0.64-1.99

 Moderately active 0.91 .655 0.59-1.40 1.09 .734 0.66-1.79

 Active 1.10 .672 0.70-1.73 1.08 .775 0.64-1.81

Self-rated general health Poor/fair/good Ref Ref 

 Very good/excellent 2.19 <.001 1.81-2.65 1.48 .001 1.18-1.84

Body Mass Index Under/Normal (<25) Ref Ref 

 Overweight (25-29) 0.73 .012 0.57-0.93 0.77 .054 0.59-1.00

 Obese (30-34) 0.43 <.001 0.33-0.56 0.48 <.001 0.35-0.65

 Very obese (35+) 0.26 <.001 0.18-0.38 0.35 <.001 0.23-0.52

Diabetic No Ref Ref 

 Yes 0.30 <.001 0.22-0.42 0.47 <.001 0.32-0.68

CKD No Ref Ref 

 Yes 0.50 .005 0.31-0.81 0.66 .131 0.39-1.13

Depression (PHQ-9) No Ref Ref 

 Yes 0.68 <.001 0.56-0.83 0.88 .288 0.69-1.11

Seen primary care doctor past year No Ref Ref 

 Yes 0.32 <.001 0.26-0.39 0.38 <.001 0.31-0.48

Reviewers' comments:

Reviewer #1: 

The aim of the study is not clearly stated. I just assume it was (1) to determine the prevalence of untreated hypertension in population samples of Arkhangelsk and Novosibirsk and (2) to identify factors to be associated with untreated hypertension in the two population samples.

Response:

The reviewer is correct and we have amended the Introduction accordingly.

Section Materials and methods should be clearly structured having the following subsections: Population (this should also include how the individuals were selected), Screening examination (should be briefly described, including blood pressure measurement), Definition of risk factors (incl. hypertension). 

Response:

We agree that the suggested structure will make the Methods section much clearer. We have now restructured and expanded the section accordingly.

The definition of hypertension used on lines 117 and 118 does not seem to include values equal to 140/90 mmHg, which is not in agreement with the quoted European guidelines on hypertension.

Response:

We have now phrased this with words, i.e. ‘at or above 140’ instead of ‘140+’. In this way it should be clearer that the guideline definition was indeed adhered to. 

Reviewer #2: 

The submitted manuscript assesses the rate of untreated hypertension based on the «Know Your Heart» study and is carried out in two Russian cities, namely Novosibirsk and Arkhangelsk. The aim was also to provide further insights into the country-specific factors contributing to the fact that far too many hypertensive men and women currently remain untreated. However, there are a number of points, which need clarifications and revision.

1) The Materials and Methods section is not enough conclusive. It is difficult to understand the methodology of the study. Although authors provided a reference to their comprehensive paper on the «Know Your Heart» study design [Cook S, Malyutina S, Kudryavtsev AV, et al. Know Your Heart: Rationale, design and conduct of a cross-sectional study of cardiovascular structure, function and risk factors in 4500 men and women aged 35-69 years from two Russian cities, 2015-18. Wellcome Open Res. 2018;3:67.], it is not enough. I recommend to give in the current paper at least basic explanations on how the study population was selected.

Response:

Following the restructuring of the Methods section as suggested by Reviewer 1, we now also include a brief overview of the sampling and study components under the heading of ‘Population’.

2) The description of statistical methods is very scarce; they are presented literally in a couple of sentences and are not allocated in a separate subsection.

Response:

In the new structure of the Methods section, we now bring together the descriptions of the statistical analysis in a new sub-section.

3) The robustness of results seems doubtful due to a low response rate. Moreover, the response rate in Novosibirsk, which is a developed city with a population of more than 1.5 million people, was more than two times lower than in Arkhangelsk (22% vs 51%, respectively). This issue must be addressed in some way.

Response:

Please see response to Editor’s comments above.

4) Within the scope of the studied problems, the psychosocial CVD risk factors such as depression, anxiety and stress are shown to be of great importance for the Russian population. These factors serve as barriers for traditional risk factors modification and CVD treatment, including hypertension treatment, and they are also known to independently affect long-term health outcomes in CVD patients. According to the design paper, the “Know Your Heart” study encompassed assessment of anxiety and depression (using validated questionnaires), as well as assessment of self-reported social support. It is not clear, why these data were not used in the multivariate logistic regression models of untreated vs treated hypertension in this paper. The inclusion of these data into the models may provide more accurate and comprehensive information on factors, associated with lack of antihypertensive treatment, specifically relevant for the Russian population.

Response:

We have included depression status (PHQ-9 instrument) in the analyses. Given that the constructs for depression, anxiety, and stress are related, we suggest that only the most severe of the three, i.e. depression, to be included. The results show that depression is borderline protective for untreated hypertension. This is consistent with the observation that hypertension is asymptomatic and less likely to be detected in individuals who are not in contact with the health system. Having other conditions, in this case, depression, may increase the probability of health system contact and treatment. While depression has been shown to affect individuals functionally in many ways, it cannot be concluded that it is a driver of untreated hypertension in the population at large.

5) I would recommend avoiding inappropriate slang expressions like antihypertensives instead of antihypertensive drugs or medicines or treatment.

Response:

We now use the term, ‘antihypertensive medications’, throughout for clarity.

---

## [Editor Report · Decision Letter 1]

13 May 2020

Untreated hypertension in Russian 35-69 year olds – a cross-sectional study

PONE-D-19-34971R1

Dear Dr. PETERSEN,

We are pleased to inform you that your manuscript has been judged scientifically suitable for publication and will be formally accepted for publication once it complies with all outstanding technical requirements.

With kind regards,

Bart Ferket

Academic Editor

PLOS ONE
---

## [Editor Report · Acceptance letter]

15 May 2020

PONE-D-19-34971R1 

Untreated hypertension in Russian 35-69 year olds – a cross-sectional study 

Dear Dr. Petersen:

I am pleased to inform you that your manuscript has been deemed suitable for publication in PLOS ONE. Congratulations! Your manuscript is now with our production department. 

With kind regards,

on behalf of

Dr. Bart Ferket 

Academic Editor

PLOS ONE